

# Can corporate supply chain sustainability standards contribute to soil protection?

Jan Frouz[1], Vojtěch Čemus[2,3], Jaroslava Frouzová[1], Alena Peterková[1,2], Vojtěch Kotecký[2]

[1] Biology Center of the Czech Academy of Sciences, České Budějovice, Czechia
[2] Charles University in Prague, Environment Centre, José Martího 407/2, Praha 6, Czechia
[3] Charles University in Prague, Faculty of Humanities, Pátkova 2137/5, Praha 8, Czechia

*Correspondence to*: Vojtěch Čemus (vojtech.cemus@czp.cuni.cz)



**Abstract.** Companies increasingly view soil degradation in their supply chains as a commercial risk. They have applied sustainability standards to manage environmental risks stemming from suppliers' farming operations. To examine the application of supply chain sustainability standards in soil protection, we combined global data on existing sustainability standards and their use in the food retail industry, a key sector in agrifood supply chains, with a case study in a medium-sized European country, to explore companies' options and views.

Soil quality is a priority objective in retail sector sustainability efforts: 41% of the investigated companies apply some soil-relevant standard. But the standards lack specific and comprehensive criteria. Compliance typically requires that farmers are aware of soil damage risks and implement some mitigation measures; however, no measurable thresholds are usually assigned. This stands in contrast to some other provisions in a number of standards, such as deforestation criteria. There are two probable causes of this difference: Companies and certification bodies have prioritised other environmental challenges (e.g., pesticide use, biodiversity loss in tropical biomes) over soil degradation. Also, there are practical constraints to the useful standardisation of soil sustainability. Effective soil sustainability provisions will require measurable, controllable, and scalable multidimensional interventions and compliance metrics. Often, these are not yet available. The development of necessary practical tools is a priority for future research. In a case study, we developed a set of standards applicable in temperate European farming practice and adapted to the needs of food retailers. Based on discussion with the industry, farmers, and soil experts, the standard is based on specific commodities rather than production units and compliance with specific agronomic practices as opposed to direct measurements of soil quality.

## 1. Introduction

### 1.1 Soils and agricultural intensification

A large majority of food used by humanity depends on soil and its ability to support plant growth (Kopittke et al., 2019). Beside food production, soils provide many other services such as detoxification, drinking water provisioning, regulation of water flow, flood protection, and climate regulation, in addition to many cultural values like heritage and cultural identity (Dominati et al., 2014). Annual value of soil ecosystem services is estimated as high as US$11.4 trillion (McBratney et al., 2017). Without exaggeration, soils are one of the most important resources economies rely upon.

Population growth has been to a large extent associated with agricultural expansion. Human population, counting about 6 million when farming emerged (Livi Bacci, 2017), has since increased dramatically. The great acceleration of the mid-20th century was supported by, among other factors, widespread application of nitrogen fertilisers (Erisman et al., 2008). At the same time, a rising proportion of people has moved into cities. As the number of urban dwellers has been increasing, the share of people working in agriculture has decreased (Satterthwaite et al., 2010; Frouz and Frouzova 2022). Moreover, affluent urban dwellers have become more demanding about food, consuming better-tasting and more expensive food, such as more meat, fat, oil, and dairy products (Satterthwaite



et al., 2010; Ericksen, 2007). Furthermore, the mean proportion of income spent on food has been decreasing with rising wealth, in accordance with Engel's law (Engel, 1857; Chai and Moneta, 2010).

Intensification and specialisation of agricultural production have contributed to these changes.

Intensification has also been accompanied with an increased influence of large food and retail companies on agricultural practices. This is particularly true for 'lead firms': global buyers who shape sales strategy, price structure, and production systems (Gereffi et al., 2005). Retailers and brand-

name food companies typically occupy this position in agrifood value chains. Retailers, processors, and traders that control a major proportion of sales often employ their bargaining power to alter trade conditions to their advantage (Ghosh and Eriksson, 2019; Fearne et al, 2005). They are also able to shape their suppliers' farm management choices. Companies' demand for high-quality produce has been linked to increased pressures on water resources, as buyers make growers follow protocols on

quality, consistency, and continuity that effectively require irrigation (Knox et al., 2010). Manufacturers' focus on ultra-processed food contributes to, for example, soil degradation (Monteiro et al., 2018). Processed food producers have been linked to significantly increased input use in agriculture (Moberg et al. 2020). Even environmentally benign practices such as integrated pest management can be driven by contractual requirements of food companies (Codron et al, 2014).


Intensification increases crop production but at the same time may often cause substantial environmental impacts (Matson et al., 1997). Agricultural intensification has been shown to reduce the biodiversity of soil organisms (Tsiafouli et al., 2015), limiting their ability to support the provision of ecosystem services (de Vries et al., 2013). Massive use of agricultural machinery enhances soil

compaction (Arvidsson and Hakansson, 1991; Kopittke et al., 2019), and together with increasing field sizes it may lead to increased erosion (Stoate et al., 2001). These effects of cultivation, together with unbalanced nutrient supply and reduced organic matter input to the soil, reduce soil organic matter content (Huggins et al., 1998). Compaction, erosion, and loss of organic matter may also feed back as decreasing soil fertility (Quiroga et al. 2006; Oldfield et al. 2019). Unbalanced nutrient use may cause

higher nutrient loss from farmland and eutrophication of water bodies, including seas (EU Nitrogen Expert Panel, 2015). Consequently, biogeochemical cycles may be affected (Kopittke et al., 2017). These effects may be further enhanced by on-going climate change, which is expected to increase the stochasticity of farm production (Tigchelaar et al., 2018). But more sustainable agricultural practices can substantially decrease these negative effects of intensification (Pretty and Bharucha 2014). In

some instances, for example, when conservation tillage or other soil-saving practices are applied, intensive agriculture may even increase removal of carbon from the atmosphere (Leahy et al., 2020).

**1.2 Soil degradation as a business risk**

Business attitudes towards the environmental impact of supply chains, including considerations of soil quality, have been changing over the past years from indifference to concern and proactive

sustainability interventions. As noted by Hajer et al. (2016), companies approach sustainability in three main ways: as a tool to improve reputation, as a sustainability-oriented business model, or through



supply chain risk management. Businesses increasingly view unsustainable practices in their supply chains as a commercial risk. Widespread soil degradation, water scarcity, and biodiversity declines are seen as potential material and, in some cases, reputational hazards. Material risks include market

volatility and potential future instability of supply chains. Market shocks facilitated by environmental change have major potential implications for costs (Tigchelaar et al. 2018). Companies fear that deterioration of natural capital may lead to direct cost increases and reduced margins, rising commodity market volatility, and supply chain unpredictability. Soil management is a risk factor due to its critical contribution to crop productivity and consequent impact on market performance (Davies,

2017; Burian et al., 2018; Panagos et al., 2018 ). Apart from primary producers and their investors, some of the most exposed sectors are the food, beverage, fibre, and biofuel industries (Makower et al. 2021). However, other, especially water-sensitive sectors are impacted as well. Climate change is expected to elevate the relative risk levels.

But companies also need to deal with other actors' concerns. The regulatory environment is increasingly stringent as governments explore effective measures to prevent soil deterioration, and damage contributes to reputational risks as well. Consumers traditionally demand a great deal from the food system: safety, quality, variety, convenience, and service as well as low prices. But they increasingly expect environmentally sustainable production and processing methods. Increasing

pressure on companies from various stakeholders such as NGOs has resulted in companies adjusting their strategies to face 'responsible governance' expectations (Fulponi, 2006, Dauvergne and Lister, 2012).

### 1.3 Sustainability standards

Government regulations and other public policies are the obvious framework that companies have
conventionally followed. However, regulations and subsidies often fail to achieve environmental needs because of weak objectives or unsatisfactory design (Frelih-Larsen et al., 2016; Paleari, 2017; Pe'er et al., 2019; Scown et al., 2020; Amundson, 2020). Since about 2000, numerous predominantly European and North American food and retail companies have sought to take a private initiative to increase the sustainability of their farm supplies beyond the minimum regulatory requirements. Initially,

their focus has been on increased sales of organic food. Organic agriculture enhances soil quality (Gattinger et al. 2012, Tuomisto et al. 2012; Henneron et al., 2014; Seitz et al., 2019), is explicitly defined, and enjoys legislative underpinning and relatively mature markets. However, its scalability remains limited. The organic share of food sales remains at around 10% in even the most advanced European markets and is substantially lower elsewhere (Willer et al., 2021). Therefore, its practical

utility as a supply chain sustainability tool is constrained.

Facing the limits of both the regulatory regime and organic segment approach, corporations have explored private pathways to mitigate environmental challenges across their supply chains. Voluntary sustainability standards (VSSs) have been a key tool. They are private norms imposed by companies

that require suppliers to follow more or less specific environmental and/or social criteria (Thorlakson et





al. 2018, Lambin et al. 2018, Traldi 2021). Suppliers' compliance with a standard is secured by a market choice to enter a private contract, as opposed to an obligatory government regulation (Henson and Humphrey 2010). Companies apply two principal approaches to VSSs: (i) third-party controlled certification schemes such as Bonsucro (sugar cane) or the Better Cotton Initiative (Vogt, 2019; Meier
et al., 2020), and (ii) in-house standards.

While companies increasingly view standards as a risk management tool, they also continue to serve as a means of responding to stakeholder expectations, communicating brand differentiation to consumers and managing business-to-business relations. They help companies to ensure product
safety or quality attributes, improve market efficiency, strengthen suppliers' liability, or induce innovation in sourcing (Fulponi, 2007; Henson 2008; Chkanikova and Lehner, 2015).

Voluntary sustainability standards are not a straightforward solution. Their geographical focus is uneven. Most of the major VSSs target tropical crops (Tayleur et al., 2017; Meier et al., 2020). They
deal with globally relevant priorities such as deforestation and biodiversity loss that are concentrated in tropical biomes, while local challenges (e.g., soil degradation), more uniformly distributed in world farming, have received less attention so far. Their real-life impact relies critically on their specific design, and some schemes may be less than efficient (Blackman and Rivera, 2011; Traldi, 2021). Research suggests a mainstreaming paradox: standard setters face a trade-off between coverage and
outcomes (Dietz and Grabs 2021). As the scope of some schemes expands beyond their original focus to cover both environmental and social agendas, parallel generalist standards overlap, their topical distinctions blur, and targeting becomes weaker (Lambin and Thorlakson, 2018). Whether this thematic generalisation impacts standards' specific content, such as environmental criteria, has not yet been sufficiently explored.

Nonetheless, VSSs are potentially an important tool of control over environmental challenges, particularly in the production of so-called soft commodities such as food and fibre. Here we investigate the extent and depth to which corporate voluntary sustainability standards are applied to protect soils, and the potential and constraints of further application of standards in soil quality. We focus on four
key research questions: (i) To what extent are companies considering soil sustainability as part of their sustainability strategy? (ii) Do sustainability standards that companies use have a potentially meaningful impact on soil protection, and does that impact affect standards' market penetration? (iii) Are schemes that emphasise the environment more likely to have stronger soil-related impact? (iv) What are companies' practical considerations in their practical application of soil protection criteria in
VSSs?

## 2. Material and methods

To explore the above-described research questions, we integrate four research approaches: (i) In order to gain an insight into the current market uptake of the relevant VSSs in business, we investigate their use in food retail, the key sector of agrifood value chains. (ii) We review the potential impact of





soil-related provisions in the existing VSSs, and (iii) examine whether it is linked to the relative
environmental specialisation of standards. Finally, (iv) we use a case study to understand food retail
companies' needs and preferences for soil-related VSSs.

**2.1 Market uptake of soil-relevant VSSs**

We investigated the application of VSSs for soil protection by global food retail. The 250 largest
retailers listed in Deloitte's *Global Powers of Retailing 2021* report (Deloitte, 2021) were used as the
baseline to determine a sample of relevant companies. Out of this sample, companies labelled
'Grocery Retailers' in the research database Passport operated by Euromonitor International were
selected in order to identify those involved in food sales. For these companies ($n$ = 119), we gathered
the latest sustainability reports, annual reports, and data from companies' websites available between
June and October 2021 and performed content analysis (Krippendorff, 1980) to identify companies'
activities in sustainable food sourcing. We focused on standards they use, crops they report as
considered in sustainable sourcing, and topics of agricultural sustainability they focus on. Using coding
based on the Sustainability Consortium's Sustainable Commodity Supply Chains Project (The
Sustainability Consortium, 2017) with some minor adjustments, we categorised relevant content
collected and removed 70 data points due to unavailability of reports and/or relevant data or language
barriers.

**2.2 Impact of soil provisions in VSSs**

Second, we analysed the content of the Standards Map (Fiorini et al., 2018), a global database of 322
VSSs (as of October 2022) operated by the International Trade Centre
(https://resources.standardsmap.org/knowledge). Out of 165 standards that cover agriculture, we
identified those that explicitly regulate soil management. We removed organic food standards
(because they are irrelevant to supplies from conventional farming) and standards focused on food
quality that only marginally mention soil. We performed content analysis of the remaining standards ($n$
= 56) and identified 11 sub-categories of criteria that the Standards Map marked as relevant to soil
(Fig. 1). We identified 400 instances where a particular standard contained one of the 11 sub-
categories. On the basis of the content analysis of the standards, we concocted four categories of
ambition level (Table 1), and assigned one to each of these individual instances in order to
differentiate between schemes with explicit benchmarks and those confined to general provisions.

We extracted from the Standards Map data on crops covered by the 56 soil-related standards to gain
an insight about the overlap between supply (existing standards) and demand (reported use by
companies for each crop). To examine whether the soil-related criteria are affected by the
mainstreaming paradox, we performed Pearson's correlation to test the relationship of the ambition
level of each individual standard to the acreage of land certified by the standard. Additionally,
Pearson's correlation was calculated to test the relationship of ambition level with the reported use of
standards among food retailers ($n$ = 18).



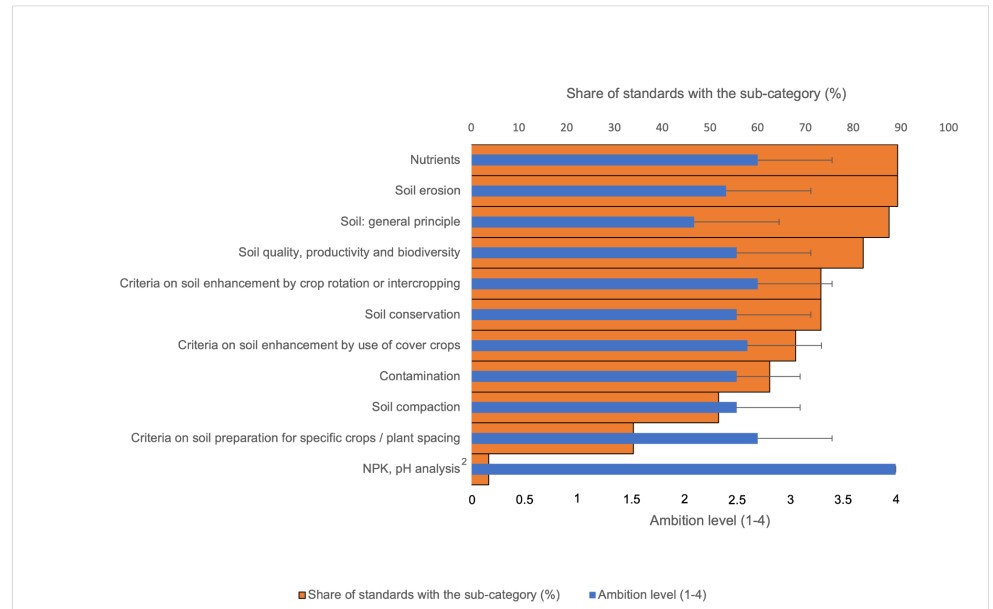

**Figure 1: Levels of supply chain sustainability standards' (*n* = 56) soil protection content ambition in individual sub-categories. Level rating criteria are explained in Table 1.**

Note:

1. Levels are applied to the sub-categories defined by Standards Map.
2. The category originally called 'Other Criteria on Soil' in the Standards Map is renamed to 'NPK, pH analysis', as this was the only actual topic covered.2.3. *Environmental specialisation*

To evaluate the environmental specialization of individual standards, we used the Standards Map (https://resources.standardsmap.org/knowledge), which indicates the proportion of requirements that are dedicated to five pillars ('Environmental', 'Social', 'Quality and management', 'Economic', and 'Ethics'). As a measure of environmental specialisation, we used the relative share of requirements in each standard dedicated to the environmental pillar extracted from the Standards Map. We applied Pearson's correlation to test the relationship between the environmental specialisation of each VSS and (i) its overall ambition level (Table 1) in soil issues; (ii) its ambition level in individual sub-categories (such as erosion, nutrients, and soil as general principle: see full list of subcategories in Fig. 1); and (iii) the area of standard application measured in hectares of certified land globally. Similarly, we compared environmental specialisation between standards that operate strictly in the tropics and/or subtropics and those that also target temperate crops. To do so we assessed the environmental specialisation of standards with these two geographic foci. The Standards Map was used to extract data about each scheme's geographical scope to differentiate between standards that regulate temperate crops (including those with a wider scope including temperate crops) and those that strictly target only tropical and/or subtropical agriculture.



| Level | Description of category | Example |
|---|---|---|
| 1 | No specific requirements or actions expected. | "If applicable, procedures are in place to measure and reduce soil erosion and compaction and/or improve soil health." <br><br> Equitable Food Initiative (Criteria on soil conservation)[1] |
| 2 | Some knowledge about agricultural sustainability issues is expected and efforts to address them are required. | "Soil Management Plan in place to avoid erosion and maintain and improve soil health Indicator" <br><br> Bonsucro (Criteria on soil nutrients)[1] |
| 3 | An explicit strategy and its demonstration in farm practices are required. | "Indicate pollution caused by the use of fertilisers and pesticides in cotton production. Applying more efficient irrigation practices to optimise water productivity (applicable to irrigated farms only)" <br><br> Better Cotton Initiative (Criteria on soil contamination)[1] |
| 4 | An explicit strategy to deal with the issue in specific measurable rules and interventions is required. | "4.1 Organic matter balance • An organic matter (OM) balance is calculated at company level. The average OM balance (balance is input minus decomposition) for all plots at company level is at least neutral. In case of a perennial crop, the balance at plot level over the entire growing period is neutral." <br><br> Planet Proof standard (Criteria on soil nutrients)[1] |

**Table 1: Standard ambition level criteria applied in the analysis**

Notes:
    1. All quotations taken from ITC (n.d.)

**2.4 Case study of standard design**

To explore the fourth research question, we conducted a case study that aimed to develop a national

standard of sustainable soil management in crop production. The case study had clearly limited scope, focussing on food retail industry in one country. However, in contrast to various questionnaire types of survey, the case study allowed us to query representatives of companies about all important steps and key decisions that have to be made during VSS preparation. The standard development involved the Czech Confederation of Commerce and Tourism, an industry association of virtually all large food

retailers active in the Czech market, a medium-sized (population 10.5 million), relatively affluent (GDP PPP US$45,700 per capita) European Union economy. We used the insights and experience gained during the development of the standard to understand the practical application, limits, and role of VSSs in real-life commercial practice. A series of facilitated workshops and semi-structured interviews with individual company managers was applied to investigate company preferences. We focused on





four critical structural dilemmas that affect implementation of standards in business operations: the
     nature of the subject entity, the operator of the standard, sustainability criteria, and geographic
     generality. Outcomes were then combined with inputs from national soil protection experts, the
     existing regulatory framework, and national good agricultural practices in order to draft soil protection
     VSSs for broader groups of crops (Novotný et al., 2017). Those were then discussed with retailers and
their suppliers for future modification. This process, which systematically aspired to acquire a
     consensus of the relevant companies, allowed the investigators to identify key design choices
     applicable across the industry.

**3. Results**

**3.1 Market uptake of soil-relevant VSSs**

Soils generally rate high among food retailers' environmental concerns (Table 2). Among the 49
     sampled retailers, 27% self-report soils as a policy objective, with only two topics – pesticides and
     water management – mentioned more frequently (both at 36%). Sustainability standards that involve
     soil protection criteria were applied by 41% of the retailers (Table 3).

Some retailers apply their own requirements, which may include both more general policies and
     specific in-house standards. Tesco operates a program within their Sustainable Farming Groups (an
     environmental initiative by Tesco involving its suppliers and farmers) that promotes use of cover crops
     and other sustainable practices in potato farming. In 2019 the program covered 417 hectares, with
     expectations to extend it further (Tesco, 2020). However, soil is generally rarely addressed in the in-
house standards. Most of them focus on pesticide use or biodiversity.

| Objective | Share of food retailers that report the objective (%) |
|---|---|
| Pesticide management | 32.7 |
| Water resource management | 32.7 |
| Biodiversity | 26.5 |
| Deforestation | 26.5 |
| Soil health | 26.5 |
| Fertiliser management | 20.4 |
| Land use change | 8.2 |
| Energy consumption | 6.1 |
| Manure management | 6.1 |
| Pollination | 6.1 |
| Ecosystem services | 4.1 |





| Habitat/land conversion | 4.1 |
|---|---|
| High conservation value areas | 4.1 |
| Maximum residue levels | 4.1 |

**Table 2: Self-reported priority agrifood sustainability objectives of 49 large retail companies**


| Standard | Share of retail companies reporting use (%) | Average ambition level | Number of sub-categories covered by the standard | Share of environmental topics in the total number of criteria (%) |
|---|---|---|---|---|
| Involves temperate crops only or in a combination with tropical/subtropical crops | | | | |
| PlanetProof | 2.04 | 4.00 | 10 | 60 |
| Red Tractor (Combinable Crops) | 4.08 | 2.20 | 5 | 56 |
| GLOBALG.A.P (Crops) | 26.53 | 2.00 | 9 | 39 |
| LEAF Marque | 6.12 | 3.00 | 10 | 71 |
| Rainforest Alliance - 2020 | 44.90 | 2.90 | 10 | 38 |
| Better Cotton Initiative | 20.41 | 2.89 | 9 | 37 |
| Sustainable Rice Platform | 2.04 | 2.67 | 6 | 47 |





| | | | | |
|---|---|---|---|---|
| Sustainably Grown | 2.04 | 2.33 | 9 | 39 |
| Round Table on Responsible Soy Association | 24.49 | 2.25 | 8 | 46 |
| Involves tropical/subtropical crops only | | | | |
| Roundtable on Sustainable Palm Oil | 59.18 | 2.63 | 8 | 34 |
| Cocoa Horizons – Barry Callebaut | 8.16 | 1.88 | 8 | 36 |
| FairTrade | 40.82 | 1.29 | 7 | 39 |
| All standards | 41 | 2.48 (median=2.33) | 7.21 (median=8.00) | 46 |

**Table 3: Average ambition level across the relevant sub-categories of standards reported as used by retailers, and the share of retailers (*n* = 49) reporting use of the standard. Level rating criteria are explained in Table 1.**

Notes:

1.    Rating is applied to the sub-categories defined by the Standards Map.

**3.2 Impact of soil provisions in VSSs**

Practical implementation of policy objectives in explicit VSSs remains limited. Just 56 of the 165 third-party standards relevant to agriculture (excluding organic certification) regulate soil management to a greater extent than only mentioning its importance. Overall, the average ambition level of the

standards' soil management requirements by sub-category (Table 1) is less than 2.48, with the median at 2.33 (Table 3); that is, they typically require that farmers are knowledgeable about soil-related risks and show some effort to apply practices to improve soil quality. The most frequent sub-categories are soil erosion, nutrients, soil biodiversity, and productivity (Fig. 1). NPK/pH analysis is the sub-category in which the standards have the most ambitious criteria overall, as compliance with exact thresholds is



required; however, it is only rarely applied (*n* = 2). There is not much variability in the level of ambition
beyond that (Fig. 1).

While there is a weak negative correlation (Pearson coefficient, *r* = −0.23, *n* = 18) between the
standard's ambition level and its hectare coverage in terms of certified production land, the
relationship is not statistically significant (*p* = 0.355), possibly due to the lack of available data (Fig. 2).
The same is the case with the relationship between the average ambition of the standard and its use
by food retailers (Pearson coefficient, *r* = −0.25, *p* = 0.441, *n* = 12). The crops most frequently covered
by VSSs are soy and fruits, both in terms of the number of standards and in reporting by the food
retailers (Fig. 3). But some standards diverge in these two criteria: for example, while a high number of
VSSs cover the sustainability of sugar, nuts or rice, they are rarely reported as used by the retail
companies.

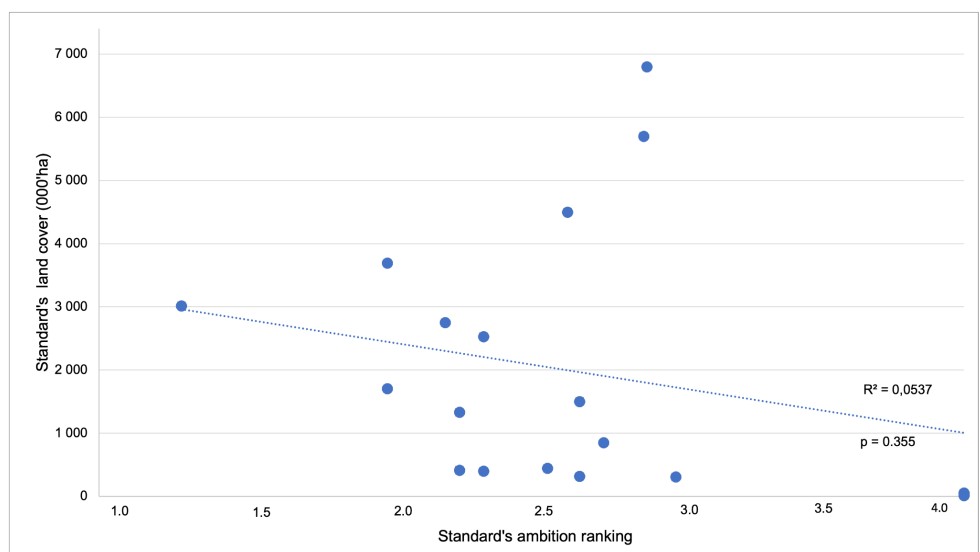

**Figure 2: Correlation between standard use measured in thousands of ha of land and standard ambition
level using available data (*n* = 18). The relationship is not statistically significant.**







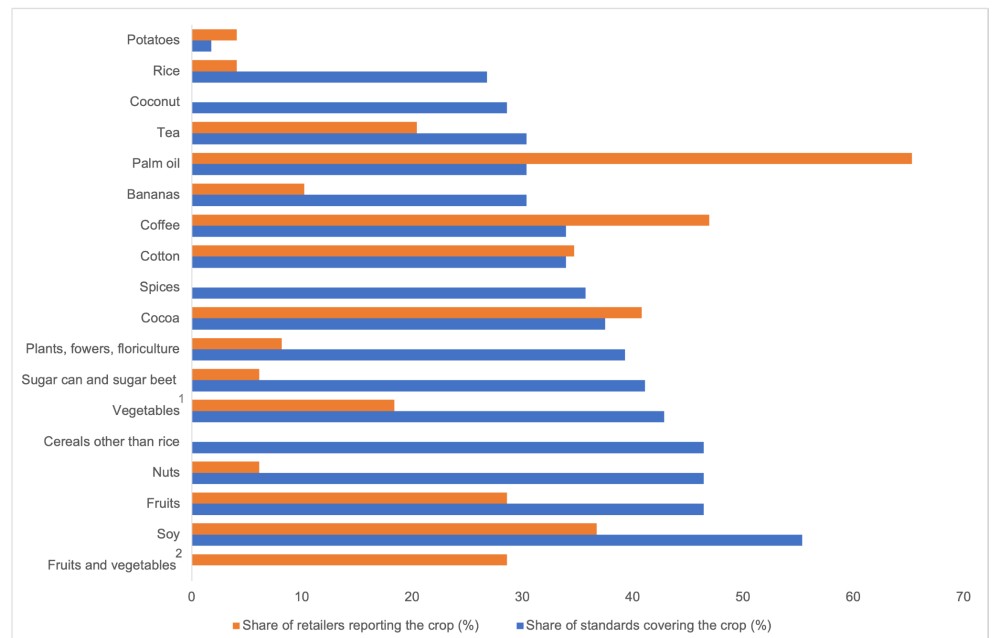

**Figure 3: Crops covered by third-party agricultural sustainability standards relevant to soil quality (*n* = 56) and those reported in food retail companies' (*n* = 49) literature as being subject to a specific sustainability standard.**

Notes:

1.  Retail companies usually report 'sugar' as a commodity, rather than the specific crop; in only one data point (1.8%) is sugar beet explicitly reported.

2.  Some companies report 'fruits and vegetables' as a generic crop category.

### 3.3 Environmental specialisation

Environmental specialisation was weakly but significantly positively correlated to average ambition level of all soil-related criteria in a given standard (Pearson coefficient, r = 0.37, *p* = 0.005, *n* = 56). There was also a positive relationship between the relative environmental specialisation of standards and their ambition levels in the erosion (Pearson coefficient, r = 0.41, *p* = 0.003, *n* = 56), soil conservation (Pearson coefficient, r = 0.32, *p* = 0.043, *n* = 56), and cover crop (Pearson coefficient, r = 0.30, *p* = 0.069, *n* = 56) sub-categories. Environmental specialisation was negatively correlated with the use of the standard measured in hectares of certified land globally (Pearson coefficient, r = −0.53, *p* = 0.025, *n* = 18); that is, standards with a stronger environmental focus are used on relatively smaller areas, and vice versa. Standards with high environmental specialisation also tend to be those operating in temperate regions, as opposed to standards that target tropical crops only (t test, *p* = 0.001, *n* = 56).

### 3.4 Case study of standard design



The authors, in cooperation with the Czech Confederation of Commerce and Tourism (an industry body of, e.g., retail companies), developed a soil protection standard to be used by the retail industry in Czechia. Standard development was guided by the ISEAL principles of aspirational, rigorous, and efficient environmental standards (ISEAL, 2013).

Based on consultations with the retail industry, a commodity-based standard was prioritised over a structure operating at the farm level. Consequently, four separate standards were developed for major crops in Central Europe: vegetables, potatoes, the other high erosion risk row crops (maize, sunflower, beet, and *Vicia faba* beans), and all other row crops (typically, cereals and oilseed rape). The standard is designed to be applied in-house in direct supplier-buyer contracts, although some shared infrastructure (e.g., a farmer registration website) is provided. The consensus of retail industry
stakeholders was that an in-house scheme is preferable to a third-party certification, due to assumed lower costs and administrative requirements.

On the basis of a consensus of retail industry stakeholders, the standard was then based on monitoring and reporting of information provided by the farmers or obtained from public databases, not
on evaluation of soil properties that would require soil sampling analysis and data interpretation, most likely by a third party. Again, here two arguments were mentioned in favour of this decision. The first was the assumption that soil analysis and third-party evaluation may require a larger additional cost. The second was the fact that future contracts would be made based on assessments of past performance that may need a longer history, while compliance with rules can be reported almost in
real time. This made the supply chain more flexible and easy to enter without a history.
The standard uses a range of interventions to achieve two measurable goals: to prevent extreme soil erosion events (defined by the national law as loss of more than 9 t.ha$^{-1}$.yr$^{-1}$ for most soils), and to achieve no net long-term loss of organic matter. The goals were developed in consultation with national experts working in the field, and chosen so that pre-existing data could be used to evaluate
their impact. A package of nine interventions was developed to achieve the goals; a different combination of these interventions was applied to each of the commodity-specific standards. All but one of the interventions control farming practices rather than soil properties. The results of this project suggest a way to standardise a range of available practices into a generic instrument, application of which will substantially reduce soil erosion and the loss of organic matter in conventional farming. The
criteria go beyond government regulation requirements (including mandatory conditions of farm subsidies, a *de facto* regulation). At the same time, they were designed so that most interventions are eligible for EU's Common Agricultural Policy agri-environmental-climate payments (AECMs) or eco-schemes, cutting farmers' costs. So it represents a kind of best practices within the framework of the already established AECMs scheme, which is the scheme the farmers are familiar with in terms of
both practical application and reporting.

Each intervention was supplemented with compliance indicators and thresholds and reporting/audit procedures described in a user manual. A great deal of effort was applied to simplify reporting so that



the administrative costs to farmers and companies were kept at a minimum, as stakeholders often
reported that the administrative burden in similar schemes tends to be more costly than the adoption
of technical requirements per se. Field-level compliance with five components can be fully verified in
pre-existing national databases; the remaining items can be easily assessed visually.

## 4. Discussion

### 4.1 Current practice

The food retail industry declares a high degree of interest in soil quality. Soil quality and/or its
individual parameters are one of the self-declared priority objectives for retail industry sustainability
efforts. However, there is an apparent discrepancy between this proclaimed prioritisation and the
implementation of any real measures into standards (Fig. 1). Soil-relevant items generally, with one
exception, lack more comprehensive and/or specific criteria. Hence, soil protection is often reported as
a priority, but practical implementation is limited. Apart from organic food, GLOBALG.AP is the most
popular standard. Soil quality is covered by the scheme, but its criteria tend to be loose and weak. In
order to qualify, suppliers must, for example, develop a crop rotation plan and implement some
interventions to mitigate soil erosion and compaction; however, no specific measures or thresholds are
required.


The explanation for the discrepancy between prioritisation and implementation is complex. Partly it is
that any evidence-based policy (Mosse 2004) needs data and data processing, and its implementation
is more complex than just the simple declaration "we care". This is particularly true for soil. Soil
sustainability criteria are also relatively more difficult to develop and control (sect. 4.2). Environmental
schemes that prioritise landscape-level threats such as land-use changes in global biodiversity
hotspots can use fairly simple metrics such as the absence of deforestation (Lambin et al. 2018,
Garrett et al. 2019). Mitigation of soil risks is typically more complex and involves field-level
interventions that are often more geographically specific. Companies may be naturally inclined to
engage first with topics that are easier to approach, measure, and verify. These complexities are
probably visible in the ways current sustainability VSSs specify soil quality requirements. While
relatively strict requirements are applied in easily verifiable measures such as use of cover crops,
crop-spacing, or soil pH, issues like soil erosion and organic matter loss are left to more vague criteria.
We will further examine the complexities and challenges faced by the development of a soil standard
in sect. 4.2.


A second problem can be that the relationship of soil to a final product is mediated by other factors,
and soil changes are usually slow, so its degradation may not be perceived as an imminent threat.
Consequently, while retail business apparently views soils as a potentially important issue, the initial
focus of its supply chain sustainability efforts has been elsewhere. Companies tend to concentrate on
major global concerns (climate, biodiversity, deforestation, and other habitat loss). This is associated
with public awareness about soil which is, despite recent efforts and some partial successes (Dazzi
and Lo Papa, 2022), lower compared with awareness of other issues such as biodiversity and climate.




There are many reasons for this. Among others, soil, soil organisms, and soil processes responsible
for soil fertility are virtually invisible to most of the population, including customers and company
managers. Thus, these matters are spotlighted less than other natural resource issues such as
biodiversity, which is easier to visualize, making it easier to build emotional attachment to biodiversity
(Hanisch et al., 2019).

The relevant agrifood supply chain impacts are generally higher in tropical and subtropical landscapes
(Moran and Kanemoto, 2017; Pendrill et al., 2019) than in temperate zones. Tropical farming is
understandably a primary priority for private schemes (Tayleur et al., 2018). These risks are also the
key priority for conservation NGOs and other stakeholders who often play a major role in companies'
understanding of sustainability agendas and their strategic choices. Reporting of the 49 large grocery
retailers shows that some of the most frequently applied schemes are the Roundtable on Sustainable
Palm Oil, the UTZ–Rainforest Alliance, and Fairtrade. These standards have one thing in common:
they mostly focus on tropical cash crops such as cocoa, coffee, and palm oil. While they typically
include some soil-related criteria, their main environmental components usually revolve around
biodiversity and habitat conversion.

**4.2 Challenges to the standardisation of soil sustainability**

**4.2.1 Sustainability criteria**

Several issues need to be considered in the standardisation of soil sustainability (Table 4). Perhaps
the first question is what should be the subject: a spatial unit (e.g., a farm) or a unit of production (e.g.,
a certain amount of a specific crop). The former approach is obviously better suited for farm-level
interventions, while the latter is easier to integrate into supply contracts.


| Decision | Options | Consequences |
|---|---|---|
| What should be the subject entity | Farm | Using an individual farm or another spatially defined unit allows combining various approaches of soil protection such as spatial organisation of fields, non-productive plots (hedgerows and similar landscape structures), and agrotechnical approaches that affect the entire farm (crop rotation, presence/absence of livestock). On the other hand, the relationship to particular commodities is less clear, especially in mixed farming regions. |
| | Commodity | Makes compliance control and traceability easier. Farmers are able to focus additional efforts on land that produces specific commodities. However, standards may interfere with crop rotation and other farm-level choices. |
| Who should operate the standard | Third-party standard | Third party schemes are more convenient for buyers as development, implementation, traceability and compliance control are fully outsourced. Also, certification bodies provide the necessary technical expertise. However, criteria are inevitably general enough to serve many users. |



| | In-house standard | In-house schemes can be tailored to particular needs and allow the buyer to design and/or operate the scheme together with its suppliers. This approach is particularly suitable for large users which use substantial amounts of commodities. |
|---|---|---|
| What should be the nature of criteria | Soil properties | Targeting a specific threshold of soil properties ensures an objective measure comparable across space and time. Farmers also get more flexibility in the way improvements will be reached. However, measuring soil properties requires specific expertise and is relatively costly; usually, it would have to be provided by third parties. Also, relevant changes are slow so that this approach would make sense only when applied long term. |
| | Farming practices | Farming practices which are presumably beneficial for soil can be relatively easy to report and monitor. Moreover, government regulations and subsidy schemes are based on compliance with certain practices. More specific requirements or elevated criteria are just a logical step in the same direction. However, the relationship between practices and soil improvements may vary based on local conditions. Also, direct responsibility for evidence gathering shifts to farmers, causing additional administrative and organisational load. |
| How geographically general the standard should be | Universal | Many large buyers in food markets are global players or use imported commodities. A universal standard is more suitable for them. However, variability of conditions and needs will be an impediment. This is one reason why many third-party international VSSs prioritise soil protection but lack specific compliance thresholds. |
| | Local (Localised) | Local or localised VSSs can be more adapted to variability of soils and other natural conditions, common agricultural practices, and existing national regulations. However, additional effort is needed to modify the scheme for local conditions. Using many local standards would increase administrative and organisational demands on users. A global framework defining basic requirements supplemented with national standards that specify them for local conditions is a possible solution. This is the approach applied, for example, by the Forest Stewardship Council, a global timber sustainability scheme (Mischkowski and Seizinger, 2016). |

**Table 4: Key choices to be considered in the development of soil sustainability standards**


The next dilemma is what the standard should control: soil quality parameters, or good agricultural practices. The obvious advantage of the soil quality approach is that the scheme will directly and measurably affect soil status. The necessary data can be obtained by periodical measurements of a

comprehensive set of soil chemical, physical, and biological properties (Baritz, 2022). This is certainly possible, but it faces technical obstacles. Soil is inherently heterogeneous, and to obtain a representative set of data characterising an entire field (or even farm) is a complex matter requiring many samples. Although this is a common problem of pedology that can be solved by advanced techniques of soil sampling, it still requires a great deal of effort and must be performed by qualified

and well-trained personnel. Sampling and analysis can strongly affect the results, and hence would





need to be done by a specialised external provider. Cost as well as technical and logistical complexities will pose a prohibitive barrier for farmers and especially companies.

Moreover, despite the efforts, the results obtained would still face considerable uncertainty. A VSS is a
contractual obligation in a business-to-business relationship. It requires clear compliance thresholds. However, minor changes in measurements may indicate either a genuine change in parameters or just random variation. This, combined with the fact that changes in many soil parameters are slow, makes it reasonable to expect that detectable improvement may appear in 5 to 10 years' time (Bartuška and Frouz, 2015). Also, some biological properties may be very dynamic in the course of a year, so one
sampling does not necessarily represent long-term values. It is possible to devise a monitoring scheme performed by an external supplier with some periodicity, perhaps 5 to 10 years, and designed to provide data for an assessment of soil quality and its change, which would have implications for farmers' future contracts. The data obtained would also enable farmers to make informed soil management choices. Nevertheless, the system would require long-term operational costs as well as
the continued attention of both buyers and farmers, and its reaction time would be slow. Reliable data about long-term trends can be obtained after three sampling intervals; that is, after at least 15 years, sharply reducing the practical utility of such a scheme to companies.

Therefore, the other option – to build the standard around a set of best practices, which the farmers
will be expected to follow – may be more practical, at least for now. Specific requirements are set so that compliance can be easily assessed. This is the approach used, for example, in the cross-compliance requirements of the EU's Common Agricultural Policy (CAP). An obvious downside is that the metrics, while easily measurable, are usually unable to capture the full variability of local soil responses to generic management practices.

**4.2.2 Variability challenges**

This raises the question of how generic a corporate soil sustainability standard should be. Companies aspire to achieve homogeneous benchmarks across diverse markets, crops, and suppliers. The criteria in existing VSSs usually follow basic objectives such as prevention of soil organic matter loss, reduced erosion and soil compaction, and improvement in soil biodiversity and its role in soil
processes. The ways to achieve them are closely related: increasing the supply of organic matter, appropriate crop rotation, minimising periods when the soil is without vegetation cover, reduced soil compaction, reduced field size, and avoidance of loss – or even restoration – of (semi-)natural habitats in the agricultural landscapes. Despite these similarities, a universal scheme with a unified set of criteria is probably implausible. The reasons for this can be grouped into three major categories: (i)
variation in environmental conditions, (ii) variability in the level of intensification and agrotechnical practices, and (iii) variation in existing national regulatory contexts.

The variability of soil conditions is well known. Few soils have a naturally high content of mineral nutrients and, at the same time, are not sensitive to degradation due to cultivation (Blum 2013, Horn et





al., 2006). Most soils are poor in mineral nutrients; some have low absorption or even bind added nutrients in forms not accessible to plants; others are highly sensitive to cultivation or have a combination of these properties. Typically, tropical soils are less fertile as they have been deeply weathered and depleted of mineral nutrients and may also be sensitive to degradation when cultivated (Blum, 2013). Moreover, other soil degradation factors such as erosion vary with climate and

topography.

The level of intensification and corresponding farming practices vary among individual countries. In the tropics, there is an on-going trend of farmland expansion at the expense of natural ecosystems. Habitat conversion is associated with soil degradation; many VSSs applied to tropical crops and

focused on (preventing) habitat loss also indirectly contribute to soil protection. In contrast, the agricultural frontier is stabilised in most temperate landscapes, with production increases achieved by rising yield per unit of area – that is, intensification. Consequently, emphasis on different indicators will depend on regional contexts. While zero deforestation is the key indicator in tropics, other parameters related to agricultural intensification are needed in temperate agriculture.


Finally, national government regulations may reflect local soil conditions, previous history of agricultural development, and past policy singularities. Regulations – which include legally binding rules and subsidy preconditions – may set specific rules for certain crops and cultivation techniques. Farmers already incorporate compliance with these rules into their operations. Any practicable VSS

needs to be compatible with regulatory contexts. However, this may also be an advantage. Individual items in the Czech soil sustainability standard are designed to leverage any existing AECMs or eco-scheme, providing farmers with an opportunity to partially or fully finance the interventions with government subsidies.

### 4.2.3 *Compliance control*

Sustainability standards in landscapes where habitat conversion is the key concern can be reasonably built around a relatively simple metric such as zero deforestation (Garrett et al., 2019). Land use changes are plausibly detectable from satellite data (Moran et al., 2020). However, monitoring of compliance with soil quality criteria – even those based on management practices such as cover crops – usually requires data collection on the ground. Checkable indicators of multiple dimensions of

compliance will be the more sensible option.

The multidimensional nature of soil damage as well as management practices also impacts companies' ability to control supplier compliance with standards. Additional reporting requirements increase the administrative burden on farmers, undermining their willingness to effectively adopt VSSs

suggested by buyers. But farmers in developed countries often already report large amounts of information to the government, typically to confirm compliance with existing regulations and subsidy



conditions, and these can be used to construct VSS metrics. However, these opportunities also constrain the construction of schemes that need to integrate the existing reporting.

### 4.2.4 Company engagement

Many of the most frequently applied sustainability standards are commodity-specific schemes (Table 3). They are typically used with commodities like cocoa or coffee that are highly visible and understandable to value chain participants as well as final consumers (Rueda et al., 2017). Also, a direct contract between the company and farmers makes business engagement in sustainable production easier. However, many row crops in particular enter diverse supply chains and levels of
processing, involving multiple tiers of traders and manufacturers. Technical specifications add to the complexity: flour, for example, requires that various types of wheat grains with different gluten levels are mixed in the milling process, weakening the link between a specific primary producer and the final product even further. Cereals and oilseeds, while crucial to the sustainability of soils in arable lands, are often difficult to recognize and regulate for downstream value chain actors. Direct engagement of
a wider range of overlapping actors on the same markets and possibly other stakeholders will make VSSs more feasible.

### 5. Conclusions

Companies' efforts to implement sustainability standards in their supply chains are a potentially important instrument of farmland soil sustainability. While companies show a rising interest in
combating market risks related to soil degradation, the practical interventions have remained in early phases so far. The efforts have been limited to some crops and issues. There seem to be several major reasons for this. Companies focus their supply chain interventions on globally important environmental risks such as loss of high-biodiversity habitats, particularly in the tropics, and more easily manageable topics such as pesticide use management. Also, soil sustainability standards
require relatively complex interventions and criteria. Provisions in the existing standards tend to be too generic to have a substantial impact.

Soils are complex, and effective sustainability standards require practical solutions that are feasible for farmers to implement and for companies to standardise, measure, and control. For example, while soil
properties are better indicators of soil quality than farming practices, the latter are often the more pragmatic choice for compliance criteria. Companies' preference for universal rules across markets is constrained by the variability of soils, farming practices, and regulatory environments. Unlike directly procured crops like fresh fruit or vegetables, complex supply chains (e.g., in processed foods) may require active engagement of a wider range of companies across markets as well as other relevant
stakeholders. Soil and sustainability research can contribute with the development of relevant tools such as multidimensional sustainability criteria, compliance metrics, and spatially explicit, commodity-relevant datasets. Some of these approaches can be reasonably applied to other complex dimensions of agrifood supply chain sustainability such as small-scale farmland biodiversity.



**Data availability**

Original research data are available on Figshare.com under DOI: 10.6084/m9.figshare.23295851

**Author contribution**

Conceptualisation, investigation, writing (lead), revision and funding acquisition: Vojtěch Kotecký.
Conceptualisation, investigation, writing and revision: Jan Frouz. Conceptualisation, investigation, data
analysis, writing and revision: Vojtěch Čemus. Investigation: Jaroslava Frouzová, Alena Peterková.

**Competing interests**

The authors declare that they have no conflict of interest.

**Acknowledgements**

This work was supported by the Technology Agency of the Czech Republic (grant number
TL03000752).

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
