# Peer review of "Can corporate supply chain sustainability standards contribute to soil protection?"

_EGUsphere, 2023_

## Author Response (AR1)

Prague, 19 March 2024

**Re: Manuscript *Can corporate supply chain sustainability standards contribute to soil protection?***

Dear Dr. Kuhn,

we thank the reviewers for their insightful and constructive input that helped to improve the manuscript.

We have edited the manuscript. Please find attached a revised version with changes highlighted.

The reviewers' concerns and suggestions are addressed in the rebuttal letter. We were pleased to use their thoughtful suggestions. The key changes we made are:

- The Conclusions are rewritten so that they more explicitly reflect the research questions, responding to the concern raised by R1;
- We discuss the different market models of the fresh and tropical crops, and their practical implications, as suggested by R2;
- A brief section on data limitations was added, directly elaborating on the issue of manufactured goods (and lack thereof in the data) raised by R2;
- A new discussion of other current initiatives, research projects and the relevant literature is included, as recommended by R2;
- We substantially expanded the discussion of other factors that contribute to companies' engagement with soil sustainability, as pointed out by R2;
- We expanded the methodology section to address the concerns of R2.
- The qualitative case study was removed, as suggested by R2.

We look forward to hearing from you.

Sincerely yours,

Vojtěch Čemus
on behalf of the authors

**Can corporate supply chain sustainability standards contribute to soil protection?**

**Authors' point-by-point response to reviewers' concerns and suggestions**

We thank both reviewers for their thoughtful and constructive input. It helped to substantially improve the manuscript. Here we summarise our responses to individual points raised in the reviewers' comments:

**Reviewer 1:**

**R1.1:** "Frouz et al. explored the effects of supply chain sustainability standards on soil protection, which focusing on four questions: (i) To what extent are companies considering soil sustainability as part of their sustainability strategy? (ii) Do sustainability standards that companies use have a potentially meaningful impact on soil protection, and does that impact affect standards' market penetration? (iii) Are schemes that emphasize the environment more likely to have a stronger soil-related impact? (iv) What are companies' practical considerations in their practical application of soil protection criteria in VSSs? The topic is important and interesting, which connects social activities and soil protection. However, after careful reading, I can't recommend publication in SOIL, as I think the four questions are not answered sufficiently, e.g. in Conclusions, I can't see the key conclusions related to the four questions. I recommend the authors have more quantitative analysis based on the data collected and clearly show if the questions are properly responded."

This is a valid and important point. While the research questions were answered in the Results, the Conclusions were rather generally worded. We expanded the Conclusions so that they explicitly and directly answer the research questions. The relevant passage of Conclusions now reads as follows:

We (i) found that the food retail industry, a key sector in agrifood supply chains, generally considers soil sustainability as part of its sustainability strategy. Sustainability standards that include soil protection criteria were applied by 41% of the sampled retail companies. However, (ii) the sustainability standards used by companies tend to have only a limited impact on soil protection. Only 56 of the 165 third-party standards relevant to conventional agriculture regulate soil management to a greater extent than simply mentioning its importance. Surprisingly, there was no significant relationship between the impact of the standard and its market penetration (hectares of certified production area). (iii) Schemes that emphasise the environment are more likely to have a greater impact on soil, particularly for criteria related to the erosion, soil conservation and cover crops.

(Please note that we also rewrote the rest of the Conclusions.)

**Reviewer 2:**

**R2.1:** „Issue 1: The paper mixes VSS on tropical products globally traded with an approach focused on local, temperate products, usually not globally traded with purchased through contracts. These are two very different situations, governed in very different ways."

This comment raises an important point. The two types of crops that appear to be prioritised by retailers (fresh fruits/vegetables and tropical bulk commodities) are indeed traded in different ways (direct purchases vs. complex supply chains), with implications for the way retailers regulate supply chain sustainability. We believe that the issue is actually even more complex, primarily for two reasons: (i) The key distinction may not be in the region of origin

and/or in the physical distance of trade flows, i.e. between tropical/globally traded and local/temperate products, but in the trade model. (ii) The market model of many of the typical temperate crops (e.g. cereals or oilseeds) resembles globally traded tropical cash crops, and it would have major implications for application of VSSs in temperate agriculture.

We addressed this issue with a brief discussion in 4.3:

Companies mostly serving European and North American markets appear to prioritise sustainable production of (i) tropical commodities and (ii) fresh produce (fruit, vegetables). They are often traded in different ways (complex global supply chains vs. direct purchases), with practical implications for implementation of supply chain sustainability (schemes such as third party certifications and direct cooperation with farmers, respectively). A meaningful intervention in soil quality in temperate landscapes would involve addressing common field crops such as cereals and oilseeds. The market model (and governance of supply chain sustainability) for many of these is more similar to that of globally traded tropical commodities, rather than fresh produce, although the physical distance of trade flows is shorter. The complexities of crops entering parallel supply chains, with supplies of different origins mixed together, and multiple tiers of manufacturers can pose challenges to the application of VSSs.

In addition, we followed this point with a broader discussion of implications for application of VSSs in soil sustainability of temperate crops:

Precompetitive initiatives (i.e. agreed and applied by several companies in a sector, potentially with involvement of other relevant stakeholders) could be a viable solution for sectoral and even cross-sectoral collaboration (Waldman et Kerr, 2014; Barker et al., 2021), enabling companies to identify best practices for their shared supply chains and focus on developing robust criteria for soil sustainability that can be measured, validated and applied interchangeably across countries and continents. Sustainable Agriculture Initiative (sect. 4.2), while not strictly a VSS, is one of the more prominent precompetitive initiatives currently on the market.

**R2.2:** "Issue 2: The paper misses out on manufactured goods that are governed by different VSS. As a result, the paper has missed the world's largest platform, SAI platform. SAI has recently developed standards for regenerative agriculture, focusing on soil health."

This is a very good point. We were aware of this. We had to choose a sector for data gathering and food retail was an apparent choice since it covers a wide range of farm commodities and plays a critical role in agrifood value chains. However, there are obvious trade-offs and less intensive reporting regarding manufactured goods by retail companies is one of them.

We addressed the issue with an additional paragraph in 4.2:

Retail industry is a natural choice of the sector for data gathering because of its key role in agrifood value chains and its broad coverage of different commodities. Nevertheless, the choice entails inevitable trade-offs. Perhaps most importantly, fresh food – a segment where they have direct contractual relationships with farmers – is an understandable priority for retail companies' supply chain sustainability efforts. As a consequence, sustainability of manufactured goods will be less intensively reported. This is, for example, probably the main reason why Sustainable Agriculture Initiative (SAI), a major collaborative platform involved in sustainability standardisation, appears in the standards data (sect. 3.2), but not in the retail data (sect. 3.1).

We also briefly return to SAI in 4.3:

Sustainable Agriculture Initiative (sect. 4.2), while not strictly a VSS, is one of the more prominent precompetitive initiatives currently on the market.

However, we do not dwell on SAI in detail, because of its scant appearance in the data.

**R2.3:** "Issue 3: The paper misses out on recent developments on environmental impact reporting in food systems (e.g., PEF) in general and on soil health indicators, MRV systems, etc., such as in France and the Netherlands (soil health index, Rabobank, Soil Heroes, Soil Capital, Earthworm, etc.) in particular. An important recent development is the EU's Mission Soil. Refer to the EJP Soil and the many Horizon Europe projects and the JRC efforts currently undertakes on these matters. Recent literature on these topics has not been used."

This is also a good point. Although often developed for other purposes, these initiatives can serve as useful tools in application for VSSs, e.g. for advanced metrics, data infrastructure etc. We added a brief discussion of those initiatives that are probably most relevant to the development of VSSs in their current form in 4.3

The relevant passage reads as follows:

The need to support soil sustainability has been the focus of many recent initiatives. In particular, the European Commission has invested significant resources in programmes such as the European Joint Programme Soil and Mission Soil, which bring together researchers, policy makers, farmers and other actors (Chenu et al., 2023) to identify priorities for soil protection (Boruvka et al., 2022) and highlight key management practices that benefit soil health (Rodrigues et al., 2021; Tiefenbacher et al., 2021; Keesstra et al., 2021; Hendricks et al., 2022; Vanino et al., 2023). Attention has also been paid to the impact of different agri-environmental schemes on soil (Polakova et al., 2022). Several EU projects have investigated incentives and business models for soil health (NOVASOIL, SoilValues, InBestSoil). Similar projects are pursued by other researchers (e.g. Soil Health Index) and businesses (Open Soil Index) (Bünemann et al. 2018). While these initiatives focus mainly on the social value of soil, public policy incentives at European, national or local level and the impact on (and support of) farmers, they produce data, monitoring infrastructure, intervention designs and other outcomes that may potentially contribute to the development of effective VSSs. Advances in agricultural mapping and remote sensing including satellite imagery will make localised soil metrics more feasible (Sharman, 2017). Moreover, with the development of AI technology, it is likely that integration of soil mapping with AI will translate into criteria and monitoring models in the future. The development of innovative monitoring, reporting and verification (MRV) methodologies to ensure the environmental integrity of carbon farming schemes generates outputs that are potentially useful for measuring other environmental impacts, including soil health (Radley et al. 2021; Springer, 2023).

We discuss the general environmental impact reporting in food systems separately in 4.3. While soil-specific initiatives focused on farm production provide important inputs into the VSSs in their current form, comprehensive LCA-based approaches (however important for e.g. reporting and labelling) are currently rather rare in VSSs. We believe that the reason is twofold: (i) VSSs have traditionally been practice-based policies, and the more recent approaches (e.g. commodity round tables) tend to follow this tradition; (ii) due to its complexity, users tend to find it difficult to apply in farm-level decision making and in contractual obligations of VSSs. This is not to say that LCA is not potentially useful for VSSs; however, the current practice emphasises other approaches.

The new text on the issue in 4.3 is as follows:

The growing breadth and depth of available life cycle assessment (LCA) data has rapidly improved our understanding of environmental footprints along agri-food value chains in recent years (Poore et Nemecek, 2018). Practical tools have been developed to apply LCA approaches at scale, such as the Product Environmental Footprint (Damiani et al. 2022). While soil quality is challenging to incorporate into LCA methodologies due to the diversity of relevant impact criteria and limited amount of soil data, numerous models and indices have been proposed (Vidal Legaz et al. 2017; De Laurentiis et al., 2019). LCA provides useful information that highlights key risk points and the relative contributions of value chain stages. As such, it is essential for reporting and labelling initiatives. Nevertheless, LCA-based criteria are rarely used in VSSs when applied to business-to-business relationships. There are probably two reasons for this. One is tradition. VSSs grew out of practice-based policies such as the organic farming standard, and more recent instruments mostly tend to follow the traditional route (Komvies and Jackson, 2013). Perhaps more importantly, LCA tends to be complex, and users (companies and especially farmers) would find it difficult to collect the necessary data and apply it to farm-level decision-making.

**R2.4:** "Issue 4: An important driver for companies is the goal to achieve net zero and the obligation to report on scope 3 emissions. The EU's CSR directive has been a major milestone in this respect, broadening the scope from carbon to other environmental issues. Refer to Deconinck et al. (2023) for a recent overview (https://doi.org/10.1093/erae/jbad018)."

We fully agree that both the net zero carbon footprint targets and the current developments in corporate environmental reporting are important drivers that contribute to the business engagement with soil sustainability. We added two relevant paragraphs in the Introduction.

The new text on carbon sequestration and its links to soil sustainability efforts reads as follows:

Along with the concerns directly related to soil sustainability, carbon sequestration is an additional motivation to intervene in soil management in supply chains. Better soil management leads to increased soil organic carbon content and is an important contribution to carbon sequestration (Smith et al., 2008; Minasny et al.; 2017; Rumpel et al., 2018; Radley et al. 2021). A growing number of companies aim for net zero greenhouse gas emissions (Hale et al., 2021; Rogelj et al., 2021). While specialist firms and initiatives such as Indigo Ag, Agreena, Soil Capital and Carboneg entered the emerging market with soil carbon credits (Popkin, 2023), many companies see working directly with their own suppliers as a useful contribution to their efforts to reduce their carbon footprint (Vermeulen et al., 2019; Amelung et al., 2020; Bossio et al., 2020).

The additional discussion of proliferation in environmental reporting is:

"Business soil conservation efforts are further facilitated by the rapid proliferation of universal sustainability reporting, propelled by regulations such as the EU's new Corporate Sustainability Reporting Directive and the expanding supply of sustainability data, tools, reporting standards and other infrastructure (Deconinck et al. 2023). Reporting contributes to agri-food companies' engagement in soil sustainability primarily by focusing their attention on the critical role of supply chains, helping them to understand their complexities and identify the less visible risks. "

**R2.5:** "Issue 5: The methodology is not well explained. More detail is needed here: what kind of coding is used in part 1, how is content analysis performed in part 2 and particularly how is researcher bias addressed when interpreting content in part 3 and finally part 4 misses any details on the case study: how many experts, what was their background, how were the workshops conducted, have interviews and workshop data be transcribed and coded, etc."

We expanded the Methodology section, specifically parts 2.1. and 2.2. as suggested. Concerning part 2.3, the data we used was extracted directly from Standards' Map , so no additional researcher's subjective assessment was involved, and the ambition level data were taken over from 2.2, so that the discussion in 2.2 is relevant here as well (we added a reference to the data from 2.2 to 2.3). The methodological issues regarding 2.4. are void since we will follow the suggestion to remove the case study entirely (see below).

The new explanation of coding in 2.1 is as follows:

Using binary coding of root word topics, based on Sustainability Consortium's Sustainable Commodity Supply Chains Project's topic classification (The Sustainability Consortium, 2017) with some minor adjustments, and related keywords, we categorised relevant content collected and removed 70 data points due to unavailability of reports and/or relevant data or language barriers. Each report was manually analysed and relevant root words recorded if they appeared; keywords (root word synonyms) were subsequently identified in the equivalent manner. Similarly, any reference to a sustainability standard was also recorded using binary coding in the data sheet. We also recorded any crop when it was mentioned in relation to a standard or a root word/keyword In this way, a binary code matrix was created, recording any instance of a root word/keyword, a standard or a relationship between any of the two variables and a crop.

The additional description of content analysis in 2.2 reads:

Content analysis often needs to go beyond simple frequency counts and involve interpretation of the text; however, these approaches increase the risk of researcher's bias (Drisko and Maschi, 2016). We used secondary data (excerpts from the Standard's Map database) and categories that allowed classification with little need of subjective judgement in order to minimise bias (Drisko and Maschi, 2016). The decision criteria were based on the presence of phrases indicating a level of ambition (Table 1).

**R2.6:** "Issue 6: Due to all the previous issues, the Czech case study contributes little to the paper. The methodology is not well explained, but also the results seem to be very limited."

Regarding issue 6, we followed the reviewer's useful suggestion and removed the case study entirely. We agree that its contribution to the paper was minimal.